# COVID-19 and Diagnostic Testing for SARS-CoV-2 by RT-qPCR—Facts and Fallacies

**DOI:** 10.3390/ijms22052459

**Published:** 2021-02-28

**Authors:** Stephen Bustin, Reinhold Mueller, Gregory Shipley, Tania Nolan

**Affiliations:** 1Medical Technology Research Centre, Anglia Ruskin University, Chelmsford CM1 1SQ, UK; tanianolan@btinternet.com; 2RM Consulting, San Diego, CA 92037, USA; reinholdmueller7@gmail.com; 3Shipley Consulting, Vancouver, WA 98682, USA; gshipley14@me.com

**Keywords:** COVID-19, SARS-CoV-2, molecular diagnostics, quantification cycle, RT-qPCR

## Abstract

Although molecular testing, and RT-qPCR in particular, has been an indispensable component in the scientific armoury targeting SARS-CoV-2, there are numerous falsehoods, misconceptions, assumptions and exaggerated expectations with regards to capability, performance and usefulness of the technology. It is essential that the true strengths and limitations, although publicised for at least twenty years, are restated in the context of the current COVID-19 epidemic. The main objective of this commentary is to address and help stop the unfounded and debilitating speculation surrounding its use.

## 1. Introduction

The sudden eruption of the coronavirus disease 2019 (COVID-19) pandemic in late 2019 caused by the novel coronavirus SARS-CoV-2 [1] has led to an extraordinary worldwide mobilisation of research, public health and governmental resources. This resulted in the rapid acquisition of extensive genetic, virological and epidemiological data that allowed the introduction of extensive testing strategies, the astonishingly swift development of a range of vaccines and therapeutics, as well as a variety of public health measures designed to contain and suppress viral transmission. There can be no doubt whatsoever that the global scientific research community’s response to this unprecedented challenge was equally unparalleled in its collective focus, immediacy and cooperation. Equally, there can be no doubt that the political, scientific and medical establishments in many countries were under-prepared and slow to grasp the significance of the pandemic, underestimated the threat it posed to the world and responded in an inadequate, delayed and often ill-advised manner. This was despite numerous reviews that analysed the stage of global preparedness for infectious disease outbreaks, identified existing gaps and proposed priority actions for strengthening outbreak prevention, detection, and response (https://apps.who.int/gpmb/assets/thematic_papers/tr-6.pdf, accessed on 20 January 2021). Incongruously, the 2019 Global Health Security Index rated the United Kingdom the best prepared country in Europe as well as ranking the country first in the world for detection, laboratory capacity and real time surveillance and reporting (https://www.ghsindex.org/country/united-kingdom/, accessed on 20 January 2021). Despite this preparation, the UK’s public health response to SARS-CoV-2 was amongst the worst in the world. There will be a time for detailed analysis of what could have been done better, and lessons will need to be learned and applied in preparation for the next pandemic. There is one narrative, however, that needs to be addressed immediately as it is an obviously false one and could lead to loss of public faith in healthcare and testing, as well as a misdirection of any forensic torchlight. This concerns the role of molecular testing, and more specifically that of the reverse transcription (RT)-polymerase chain reaction (PCR) as an essential diagnostic tool for monitoring and managing the current COVID-19 pandemic [2].

## 2. Molecular Testing

This pandemic has provided a comprehensive demonstration of the key role played by diagnostic testing in disease outbreak monitoring and control [3]. Fast, reliable and accurate testing for SARS-CoV-2 is arguably the principal prerequisite for helping to prevent the spread of COVID-19, which has presented the scientific community with unprecedented challenges. Not only is this disease caused by a novel virus, but a high percentage of all infections are transmitted by pre- or asymptomatic individuals [4,5], making the ability to identify infected, presymptomatic individuals an exigent necessity. Of equal importance, whilst the mere presence of viral RNA is not necessarily indicative of whether an individual can still transmit the virus to others [6] it is also probable that higher viral load is associated with increased disease severity and mortality [7] and, indeed, that viral load at admission independently predicts mortality [8]. Hence the ideal testing regimen would involve not just qualitative detection of SARS-CoV-2 but reliable and meaningful quantitative reporting of viral loads.

Unfortunately, in common with many other sectors, the global escalation of COVID-19 has been accompanied by a wealth of misinformation, outright lies, tendentious reporting and circulation of malicious “alternative facts”, starting with doubts about its origins, continuing with half-truths regarding the dependability of testing and extending to the efficacy and safety of therapies and vaccines. There is particular emphasis on depicting assays based on the RT-qPCR, the most commonly used method for the detection of SARS-CoV-2, as scientifically and technically flawed. In the UK, and probably elsewhere, there is a concerted effort underway to undermine confidence in the tests themselves, assail the integrity of individuals advocating their use and misrepresent their role in political decision-making. The main thrusts of these assertions allege that PCR testing is not fit for purpose, is unsuitable for large scale diagnostic testing and that there is a lack of thorough quality control. Together, these arguments are used to allege that the tests are unreliable, lack internal consistency and that they were “never meant to be used on ‘well’ people with no clinical symptoms”. Without any credible evidence, this is then used to intimate that governments have been negligent in adopting a policy of implementing PCR-based tests. These claimants insinuate that inappropriate PCR testing has prompted unnecessary lockdowns, with untold adverse economic and mental health consequences, and that the use of testing data to institute mass vaccination drive is wrong. Activities are focusing on casting a net as widely as possible, inviting the public to provide statements claiming that they, their children, or business was “adversely affected by lockdowns, quarantines or forced self-isolation based on PCR testing” for the purpose of launching “the biggest lawsuit against the government ever seen in this country” (https://pcrclaims.co.uk/, accessed on 20 January 2021). The purpose of this brief opinion is to set the record straight with regards to RT-qPCR testing.

## 3. RT-qPCR

RT-qPCR has revolutionised the diagnosis of infectious diseases. The sensitivity and specificity of the technique has allowed it to identify active or recent infection by detecting the genomes of disease-associated pathogens, including viruses [9]. The technology is highly adaptable and is relatively simple to apply. Rapid protocols have been developed that have made it an outstandingly useful and ubiquitous research tool, when used appropriately [10,11]. At the same time, unquestionably, there are issues with a test as sensitive and as powerful as RT-qPCR. Certainly the label “gold standard” is ill-advised, as not only are there are numerous different assays, protocols, reagents, instruments and result analysis methods in use, but there are currently no certified quantification standards, RNA extraction and inhibition controls, or standardised reporting procedures. In practice, therefore, the reliability of RT-qPCR results depends on a number of parameters that include sample collection and processing, the method of RNA extraction, choice of reverse transcriptase and Taq polymerase, type of probe, efficiency of assay, choice of instrument, analysis method as well as operator intervention [12]. The experience of RT-qPCR being misused as part of the measles virus/MMR vaccination/autism distortion campaign [13] has taught us that the proper utilisation of this technology requires expertise, care and transparency, with a clear emphasis on facts.

RT-qPCR unquestionably provides the most reliable, rapid, sensitive, specific and flexible means of detecting SARS-CoV-2 RNA. From the outset it is important to differentiate between the diagnostic use of RT-qPCR as a qualitative or quantitative test. The quantitative test claims to provide information regarding viral load, whereas when applied qualitatively, the presence or absence of a particular pathogen is reported. A qualitative test is usually an end-point determination although when used with target-specific fluorescent probes, the real-time amplification plots are helpful to ascertain the quality of the reaction. However, the widespread use of RT-qPCR as a quantitative research tool has resulted in its extension to quantitative diagnostic usage, where results are interpreted in terms of viral load and infectiousness. This fundamentally alters the substance of the technique, because accurate and clinically relevant quantification requires a range of controls and standards, as well as the incorporation of the data within a complementary clinical context.

There has been a long running and consistent drive to inform researchers of the challenges associated with the quantitative use of qPCR, especially when applied to RNA [14,15,16,17,18,19,20,21,22].There can be no doubt that PCR-based tests must be judiciously designed, carefully optimised and extensively validated before use, be that as research or as diagnostic tools [23,24,25]. Indeed, the authors have driven the establishment of PCR guidelines for the research community, leading and taking part in numerous workshops and publishing papers aimed at maximising the reliability of PCR-based results [26,27,28,29,30,31]. This enterprise has met with a mixed response, with many research publications failing to meet acceptable standards of reporting, appropriate experimental design, protocols or data analysis [32,33,34,35,36]. Nevertheless, there can be no doubt that an assiduously designed RT-qPCR assay, carried out with appropriate controls, standards, reference genes and analysis processes, can generate biologically meaningful quantitative data that must, crucially, include an assessment of measurement uncertainty [37].

However, publishing research papers is a world away from developing diagnostic assays, yet the combination of exquisite specificity and extraordinary sensitivity has made PCR-based methods the yardstick technologies for diagnostic testing for pathogens [38,39,40,41]. This has led to a proliferation of benchmarked kits for the detection of bacterial, fungal and viral pathogens that crucially and, unlike most assays published by research laboratories, have undergone proper validation, rigorous quality controls and CE marking or FDA approval. Consequently, far from being not fit for purpose, PCR testing was the obvious choice for detecting SARs-CoV-2, once the viral genome had been sequenced and published.

## 4. SARS-CoV-2

The SARS-CoV-2 sequence was published on January 10th, 2020, the first primers and probes targeting the virus were described on January 13th and the first SARS-CoV-2 detection assay was published on January 23rd [42]. This test, targeting three SARS-CoV-2 genes, was not ideal as primer and probes were not optimised and it was not as sensitive as later iterations. However, the assay worked and was specific and demonstrated astounding sagacity and selflessness by the scientists involved, as well as the remarkable speed with which PCR-based tests can be developed and put into practice. Since then a large number of additional tests and improved protocols have been created by individual research groups and companies in many countries, targeting a range of SARS-CoV-2 specific sequences, some detecting multiple targets, others able to detect its presence in crude samples, and yet others generating results in minutes [43,44,45,46,47,48,49,50,51]. This means that we have a whole panoply of kits available using different reverse transcriptases, Taq polymerases and buffer combinations, all of which can detect SARS-CoV-2. The performance of different tests varies of course with regards to reproducibility and accuracy and how rapidly they return results [45,52,53,54,55,56,57], but crucially, they are sufficiently sensitive to detect the virus in pre-symptomatic individuals harbouring a low viral load. Patently, PCR testing is highly suitable for large scale testing, as demonstrated daily by the millions of tests carried out to date. The acceleration of technology and expertise witnessed during the course of this pandemic is resulting in astonishing advances in convenience, speed and reach of PCR-based devices.

## 5. Challenges

Nevertheless, technical challenges and interpretative hurdles associated with SARS-CoV-2 detection have been evident for a long time. In the three paragraphs below, we highlight two of the main technical issues as well as a problem based on the interpretation of the data.

RNA quantity, quality and integrity have major bearings on the reliability of any RT-PCR-based test [58,59]. Patient samples may be obtained by qualified health professionals but are also frequently taken by inexperienced individuals or even from self-testing procedures. Sampling generally occurs at sites remote from the testing laboratories using a fairly invasive sampling technique when correctly administered. The quality of RNA extracted from samples depends on an optimised pre-test process that requires a consistent standing operating procedure with regards to the site from where the sample is collected, e.g. nasopharyngeal swab, the timing, i.e., was it collected before or after symptoms appear, who collected the sample, storage time and condition, as well as how it was processed [48,49,60,61,62,63,64,65,66,67]. Consequently, inconsistent pre-test workflows can result in variable sample make-up, with inappropriate handling of samples resulting in no detectable target being present and the RT-qPCR returning a negative result [68]. False-negative results, which change over the course of an infection [69], are problematic because they not only lead to an underestimate of COVID-19 incidence but, perhaps more acutely, will lead to infectious individuals remaining as a source of infection in the community and undermine the effectiveness of infection control measures. This is of particular concern when negative test results are used as a release mechanism for returning to work or travel. To avoid false negatives, it is essential to develop, and include, reference materials that can be spiked into patient samples to monitor RNA extraction efficiency, quality and integrity. Ideally, and in order to guarantee measurement traceability, these would be developed by a collaboration of national metrology institutes, ideally in the form of certified standards that would permit accurate and clinically relevant patient evaluation based on a combination of molecular testing results and clinical data.

The flip side concerns the potential for contamination, which will result in the reporting of false-positive results caused by inadvertent contamination of samples. For MMR/autism this was clearly established during the court-ordered evaluation of RT-qPCR experiments, where a third of negative controls were found to return a positive signal [13,70]. This has re-emerged as a problem during the current pandemic, where reagent contamination, sub-optimal PCR conditions or inadequate laboratory process control practices can quickly lead to major problems associated with amplicon contamination across laboratories and workers [71,72,73,74,75]. Furthermore, there is additional scope for environmental contamination in hospital settings, caused by extensive viral RNA-laden droplets and aerosols contaminating surfaces and airflow across a range of acute healthcare settings [76,77,78] during the collection, extraction or amplification workflow, with post-PCR amplification products always an additional potential source of contamination. False-positive results are a particular problem when prevalence is low, and the potential for adverse consequences includes unnecessary contact tracing, inappropriate treatment, removal of key workers and delayed or impeded appropriate medical treatment [79]. A checklist detailing how to tackle this problem by physically separating extraction- and PCR-related tasks, carrying out routine contamination controls of pipettes, water, oligonucleotides, master mixes and including multiple negative template and RT controls has been published [71].

Although reporting of SARS-CoV-2 test results as “detected” or “not detected” is sufficient for diagnosis, the clinical challenge arises when there is a demand to utilise the quantitative potential of qPCR results to determine SARS-CoV-2 viral load [80]. There are two significant challenges that must be addressed before the quantitative potential of qPCR results can be utilised. First, current evidence concerning viral load measurements suggests a significant degree of inconsistency very much dependent on sample origin: data from upper respiratory tract samples indicate that viral numbers peak around symptom onset or a few days thereafter, and persist for about two weeks, whereas data from sputum samples indicate higher and later peaks as well as longer persistence [81]. Furthermore, although diagnostic tests are performed on suspensions from nasopharyngeal swabs, results are intrinsically variable, dependent on the operator and on the ability of the patient to tolerate the procedure [82]. Finally, not only is it unclear what the human infectious dose is [83], an issue complicated by the emergence of more infectious new variants, but in the absence of a reference mass or volume unit, the concept of viral load itself is uncertain [84]. It is also important to note that there is currently no standard measure of viral load in clinical samples and that there is an obvious need to include a validated reference marker with diagnostic assays to make different workflows, protocols and assays comparable [85].

Second, RT-qPCR results are traditionally recorded as quantification cycle (Cq) values, which denote the number of cycles at which a qPCR instrument first detects amplified target. Cq values are adopted as surrogate indicators of the amount of target nucleic acid present in a sample [86]. The challenge is to interpret test results that are ostensibly quantitative and are reported as unambiguous quantification cycles within the clinical context of whether an individual harbours live virus and is infectious or not. The WHO Working Group on the Clinical Characterisation and Management of COVID-19 infection have recommended that measuring Cqs can be used to quantify viral burden, suggesting that whilst Cqs provide no insight into the clinical status of the patient, they can be used to measure pathogen burden in response to treatment [87]. However, this recommendation is rather unsound, as the recommendations do not specify how Cqs should be acquired or standardised. Cqs are not an objective measure as they are not standardised across platforms, reagents or analysis procedures. Indeed, Cq values can vary significantly in the same sample when different genes are targeted on the SARS-CoV-2 genome [88] and there is one strand of opinion that advises that Cq values should not be routinely reported [84]. An early US study used the CDC testing protocol to show that out of 48 individuals in a care home testing positive, 27 were pre/asymptomatic, using a Cq cutoff below 40 to indicate a positive result [89]. A week later, 24/27 developed symptoms and 3/27 remaining asymptomatic. The Cqs of all individuals were similarly high at 21-25 regardless of whether they had symptoms initially and the symptom onset date did not appear to be associated with Cq increases or decreases. The study used both nasopharyngeal and oropharyngeal swabs that were analysed separately, but experimental protocols and details descriptions of the results are sketchy, at best, and there is no mention of extraction or inhibition controls having been used.

We have argued for a long time that, far from being unambiguous, Cqs can conceal as much as they reveal in the absence of standardised analysis and reporting procedures [15,16] an issue considered as one of the key elements of the Minimum Information for Publication of Quantitative Real-Time PCR Experiments (MIQE) guidelines [27]. Crucially, the use of arbitrary Cq cutoffs is inappropriate, because they may be either too low, thus eliminating valid results or too high, thus increasing false-positive results [90]. The practical implications of this uncertainty are illustrated by a recent court case in Portugal, which used the ambiguities surrounding the clinical relevance of high Cq values [91] to rule that, given the lack of information concerning PCR analytical parameters, such tests are unreliable. Some of these issues are illustrated in the data shown in the two figures Appendix A.

The challenge to link Cq values and prognosis is also exemplified by some of the results reported in the literature, where the association of low Cq values and worse outcome has led to contradictory conclusions. Whilst some found Cq to be associated with [7] or an independent predictor of disease severity [92] and patient mortality [93,94], at least one study concluded that viral load was a poor predictor of disease outcome [95]. There is no consistent or reliable protocol for determining Cqs amongst these, and many other, published papers. Some use standard curves, for example with the US CDC 2019-nCoV_N1 primers and probe set, to determine viral copy numbers, others simply record Cqs values without reporting what the selection criteria were.

RT variability and PCR efficiencies are rarely reported, and it is quite obvious that basing clinical decision-making and health management policies based on raw quantification cycles is inadvisable. This was illustrated by an external quality assessment of 66 laboratories incorporating a range of extraction, RT platforms and reagents, which found that nearly 8% of reported Cqs derived from identical samples deviated by more than four cycles, with a maximum of 18 Cqs. The authors concluded that in the absence of standardisation, Cqs do not meet the standard of a diagnostic test when estimating viral loads [96]. These results mirror those reported in an earlier evaluation of eight clinically relevant viral targets in 23 different laboratories, which resulted in Cq ranges of more than 20, indicative of an apparently million-fold difference in viral load in the same sample [97].

Taken together, these data strongly argue the case for standardisation and availability of calibration controls to obtain reproducible Cqs, which must then be combined with large scale test results and correlation with patient data to be translated into clinically useful information. It is certainly clear that (i) whilst samples recording high Cqs are indicative of low amounts of RNA, simple reporting of Cqs is not a suitable indicator for clinical viral load, (ii) reliance on a single, raw Cq value is wholly inappropriate, (iii) using Cq values as cutoff points is meaningless and (iv) the association between higher Cqs and infectiousness is unclear.

## 6. Conclusions

There can be no doubt that, in principle, RT-PCR in general and RT-qPCR in particular, are technologies eminently suitable for fast, accurate, reliable and high throughput molecular diagnostic testing, especially when employed creatively, for example by pooling samples [98]. RT-qPCR has proven its proficiency by having been the benchmark diagnostic technology for the identification of numerous pathogens, including viruses, for many years. However, despite the remarkable swiftness with which scientists have used sequence data to design SARS-CoV-2-specific assays, the public health management of RT-qPCR testing programmes in many countries has been wholly inadequate, poorly organised and consequently, frequently ineffective [2]. This has resulted in widespread confusion, misperceptions and misinterpretation of RT-qPCR testing results [99], to the point where the undisputable uncertainties surrounding the relevance of detecting low levels of viral RNA have been hijacked by an assortment of special interest groups. Coupled to the swiftness and breadth of the transmission of COVID-19, this has led to significant challenges with regards to the understanding of those results, especially when they are pertaining to evaluating the clinical relevance of very low viral loads. Clearly, whilst just over one year after the first emergence of this novel virus, RT-qPCR testing continues to be a key tool for the reliable, sensitive and specific detection of SARS-CoV-2, there are a number of points that must be tackled to ensure that this technology performs at its optimal level:The pre-analysis steps that comprise sample collection, storage, transport and processing must be standardised and optimised, since the best RT-qPCR assay cannot perform adequately on a sub-standard sample.Ideally, kit manufacturers should release the sequences of both primers and probes.As with any other diagnostic test, manufacturers must continuously monitor and validate assay designs and reagents to ensure they remain fit for purpose. It is particularly incumbent on them to constantly screen newly emerging variants for mutations in the binding sites targeted by their assays.Manufacturers’+++ protocols are mired in the past, with overlong RT, denaturation and polymerisation times [100]. A collective aim might be to introduce “extreme” PCR conditions that would exploit the full speed potential of the latest Taq polymerases and result in assay times of below a minute [101,102]Testing sites must follow protocols punctiliously with particular emphasis on the prevention of contamination.The evident lack of certified standards or even validated controls to allow for a correlation between RT-qPCR data and clinical meaning requires urgent attention from national standards and metrology organisations, preferably as a world-wide coordinated effort.In the absence of such standards, whilst lower Cq-values generally correlate with higher levels of viral RNA, the quantification and precision associated with differences in Cq-values, especially at levels close to the limits of detection, have not been established. As a consequence, the concept of “high Cq” is vague and hence ambiguous, its translation into a clinically valid assessment of infectivity remains a matter for discussion and results must be interpreted judiciously.

Fortunately, given time, there can be little doubt that the development of standardised workflows and optimised protocols, together with appropriate certified reference materials and an understanding of the relationship between Cq, viral load, infectiousness and prognosis will allow clinicians, epidemiologists, virologists and ultimately politicians to prudently and knowledgeably make decisions for individual patients. Finally, it is important to explore the potential of alternative testing procedures to complement molecular and serological testing procedures. These include CT scans [103], which are sensitive as well as informative if not easily suited to mass screening or COVID-19-specific white blood cell “signatures” that could help distinguish SARS-CoV-2 infection from community-acquired pneumonia in patients presenting with pneumonia symptoms [104].

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
