# Peer review of "COVID-19 and Diagnostic Testing for SARS-CoV-2 by RT-qPCR—Facts and Fallacies"

_ijms, 2021, doi:10.3390/ijms22052459_

Round 1
Reviewer 1 Report
Bustin’s report entitled “COVID-19 and diagnostic testing for SARS-CoV-2 by RT-qPCR – facts and fallacies” demonstrates the true strengths and limitations of the RT-qPCR-based diagnostic technology by comparing different master mixtures in different instruments. The manuscript is convincingly and logically described. This commentary could guide clinicians and companies to be more prudent and scientific when they have to make diagnostic decisions on SARS-CoV-2 infection or therapeutic treatments or when they manufacture diagnostic kits.
Minor points:
1) In Fig. 1A, it is recommended to describe sequence or kit information of primers and probes, MM1 to MM7, that they used.
2) In Fig. 2B, different RNA samples should be marked on the graph.
Author Response
Thank you for your comments.
- We have clarified in the Figure legend that all primer and probe sequences were shown in our earlier paper published in December (reference 46).
- 3. We feel its would be unfair to rank the master mixes by supplier, as result could be different with different assays. Furthermore, we have noticed in the past that when we did publish which master mix corresponded to which suppliers that the supplier used this to advertise the "superiority" of their reagents.
- We are not quite clear what we should mark, as the RNA samples are clearly described as samples 1 and 2. But be re-emphasise this in the legend.
Reviewer 2 Report
The authors brilliantly report current findings in SARS-COV-2 PCR testing, giving clear explanations to false claims that are actually widespread also within the scientific community.
Despite the paper is well written, I suggest adding in the text, were the authors think is appropriate the following points:
- Explaining the actual risks for a false negative result (i.e. a falsely negative patient is admitted to a non-covid ward or is free to roam in the outer environment)
- What can be done to decrease the rate of false negatives (imaging?)
- Report that some authors have tried to develop other non-invasive tests to detect COVID-19 in patients with symptoms based on WBC count (I deeply suggest to cite this paper published in another mdpi journal: https://doi.org/10.3390/diagnostics10090619) and the authors view on the topic.
Author Response
Thank you.
- We have added comments concerning false negative results and referred to and referenced CT imaging. However, since this is a dissection of RT-qPCR, we felt it was inappropriate to discuss these issues any further.
- We have included a reference to the suggested potential use of white blood cell counts at the end.